# In Vivo Quantification of the Effectiveness of Topical Low-Dose Photodynamic Therapy in Wound Healing Using Two-Photon Microscopy

**DOI:** 10.3390/pharmaceutics14020287

**Published:** 2022-01-26

**Authors:** Hala Zuhayri, Viktor V. Nikolaev, Anastasia I. Knyazkova, Tatiana B. Lepekhina, Natalya A. Krivova, Valery V. Tuchin, Yury V. Kistenev

**Affiliations:** 1Laboratory of Laser Molecular Imaging and Machine Learning, Tomsk State University, 36 Lenin Av., 634050 Tomsk, Russia; zuhayri.n.hala@gmail.com (H.Z.); vik-nikol@bk.ru (V.V.N.); a_knyazkova@bk.ru (A.I.K.); tatiana_lepekhina@mail.ru (T.B.L.); nakri@res.tsu.ru (N.A.K.); tuchinvv@mail.ru (V.V.T.); 2Science Medical Center, Saratov State University, 83 Astrakhanskaya Str., 410012 Saratov, Russia

**Keywords:** wound healing, low-dose photodynamic therapy, photosensitizer, two-photon microscopy, second-harmonic-generation-to-auto-fluorescence aging index of the dermis

## Abstract

The effect of low-dose photodynamic therapy on in vivo wound healing with topical application of 5-aminolevulinic acid and methylene blue was investigated using an animal model for two laser radiation doses (1 and 4 J/cm^2^). A second-harmonic-generation-to-auto-fluorescence aging index of the dermis (SAAID) was analyzed by two-photon microscopy. SAAID measured at 60–80 μm depths was shown to be a suitable quantitative parameter to monitor wound healing. A comparison of SAAID in healthy and wound tissues during phototherapy showed that both light doses were effective for wound healing; however, healing was better at a dose of 4 J/cm^2^.

## 1. Introduction

Wound healing is a complex physiological and dynamic process that occurs in the skin at the cellular level. It involves different overlapping phases of cellular activity that occur in the proper sequence, at a definite time, and for a specific duration [1,2]. Hemostasis begins within minutes of a wound occurring with vascular constriction and ends in the formation of a fibrin clot. The latter is considered essential in promoting the onset of the inflammatory and repair phases [3]. As soon as the clot is formed, pro-inflammatory cytokines and growth factors are released, such as fibroblasts, platelet-derived growth factor and epidermal growth factor. Inflammatory cells migrate into the wound and promote the inflammatory phase, characterized by the sequential infiltration of neutrophils, macrophages, and lymphocytes [3,4]. Neutrophils play an important role in clearing microbes and cellular debris in the wound area. Macrophages are responsible for inducing and removing apoptotic cells, thus paving the way for the resolution of inflammation. As macrophages clear apoptotic cells, they undergo a phenotypic transition to a reparative state that stimulates keratinocytes, fibroblasts, and angiogenesis to promote tissue regeneration. In this way, macrophages promote a transition to the proliferative phase of healing [5]. The latter generally follows and overlaps with the inflammatory phase, beginning approximately on the third or fourth day after injury. It includes granulation tissue formation, re-epithelialization, neoangiogenesis, regeneration of the extracellular matrix (ECM), and wound contraction [6,7].

Within the wound bed, fibroblasts produce collagen, glycosaminoglycans, and proteoglycans, major ECM components. Following proliferation and ECM synthesis, wound healing enters the most prolonged and final remodeling phase, typically beginning one week after injury following collagen deposition in the wound [8]. Collagen, including collagen types I and III, is one of the major skin components [9]. During the wound healing process, type III collagen is replaced by type I collagen, which provides tensile stiffness. Collagen fibers remodel by aligning with the body’s tension lines and gaining strength through cross-linking [7,8,9,10]. In this final stage of healing, an attempt to recover the normal tissue structure occurs, and the granulation tissue is gradually remodeled, forming scar tissue [10,11].

Photo processes induced by external radiation mediated by photoactive compounds in tissues contribute to skin disease curing, skin rejuvenation, and wound healing [12,13]. Photodynamic therapy (PDT) is widely used to treat skin diseases like acne, viral warts, and skin cancers [14,15,16,17]. PDT enables a reduction in treatment time, accelerated tissue repair, and the promotion of wound healing [18,19].

PDT is based on using a photosensitizer (PS), which is accumulated in tissues, followed by irradiation of the tissue with a light source of an appropriate wavelength. The latter causes the generation of reactive oxygen species (ROS) [14,20]. ROS stimulates various cell administration pathways: apoptosis, necrosis, autophagy, depending on the type of cell treated, the concentration of the PS used, the dose of light energy supplied, and PS intracellular location [20,21,22]. High concentrations of ROS cause cell death, while low concentrations can cause the triggering of cellular repair processes, including proliferation and autophagy, and offer treatment by promoting healing [20].

The first PS generation was mainly constituted by hematoporphyrin and its derivatives, which have low selectivity and high cutaneous phototoxicity [23,24]. The second PS generation included porphyrin, phthalocyanine, benzoporphyrin, thiopurine derivatives, chlorin, and phenothiazines. It is characterized by greater deep light penetration capability due to its maximum absorption in the 630–800 nm spectral range corresponding to the tissue’s highest transparency, good solubility, and higher chemical purity [14,25]. For skin treatments, 5-aminolevulinic acid (5-ALA), which is converted into the endogenous PS protoporphyrin IX (PpIX), has an absorption maximum at 630 nm and is widely used [22,26,27]. Methylene blue (MB) is a well-known phenothiazine derivative with great importance in medicine, including antimicrobial and antiviral light-induced activity [25,28,29]. The third PS generation corresponds to enhanced forms of the first and second generations. It is often combined with a carrier such as a lipoprotein, liposomes, nanoparticles, or conjugated antibodies to increase selectivity and contrast in affected areas [30,31,32,33].

According to the Arndt–Schultz law, no tissue response will occur when low-level laser light is applied with a low dose. If used with a high dose, it can inhibit tissue response and even induce the proliferation of cancer cells or microorganisms [34,35]. Therefore, there is an optimal dose where a maximal response is obtained [36]. Some studies suggest that open wound healing stimulation occurs in the 0.5–1 J/cm^2^ light dose ranges and the 2–4 J/cm^2^ range for superficial wound healing stimulation through the skin [37,38]. Alternatively, doses were proposed in the region of 4 J/cm^2^ with a range of 1–10 J/cm^2^ for superficial targets [38]. However, according to most previous studies, the optimal light doses are in the 1–5 J/cm^2^ interval [39,40,41]. Byrnes et al. established that 632 nm light at an energy density of 4 J/cm^2^ activates collagen formation [41]. Prabhu et al. studied the effect of a 632.8 nm laser with different energy doses, including 1 J/cm^2^ and 2 J/cm^2^, and achieved good results [42]. Another study demonstrates the utility of photobiomodulation (PBM) therapy (810 nm with a total energy of 3 J/cm^2^) in mitigating burn injury and provides the biological rationale for its clinical application in wound healing [43]. PBM with various light parameters has been used widely in skincare but can cause certain types of unwanted cells to proliferate in the skin; this can lead to skin tumors, such as papillomas and cancers. H. Goo and his colleagues confirmed that LEDs with a wavelength of 642 nm with a total fluence of 21.6 J/cm^2^, increased tumor size, epidermal thickness, and systemic proinflammatory cytokine levels [34]. In an infrared neural stimulation (INS) study, Throckmorton and colleagues evaluated the parameters of light such as wavelength, radiant exposure, and optical spot size using three commonly used wavelengths of INS, 1450 nm, 1875 nm, and 2120 nm. The pulsed diode lasers at 1450 nm and 1875 nm had a consistently higher (∼1.0 J/cm^2^) stimulation threshold than that of the Ho:YAG laser at 2120 nm (∼0.7 J/cm^2^). An acute histological evaluation of diode-irradiated nerves revealed a safe range of radiant exposures for stimulation [44].

Since two optical methods are mainly used for wound healing—PDT based on exogenous PSs and low-level light therapy (LLLT) without the use of exogenous PSs—it is crucial to find an optimal technology that combines the advantages of both methods—using low doses of radiation, correcting the effects that stimulate healing wounds, and using selective staining of wounds with exogenous photosensitizers with an optimal (relatively low) concentration. This approach is called low-dose PDT (LDPDT).

Traditionally, wounds are observed invasively with a histochemical assessment of biopsies. Similar methods are non-quantitative and may cause tissue damage and healing delay [45]. Recent studies have turned to optical imaging methods. Two-photon microscopy (TPM) is a modern molecular imaging method that enables noninvasive evaluation and monitoring of skin morphological structure and functions at the cellular and subcellular levels with a high spatial resolution [46,47,48]. TPM, including autofluorescence (AF) and second harmonic generation (SHG), can provide functional and structural imaging of biological tissue and assess cells’ in vivo metabolic status [48]. Type I collagen induces SHG due to its noncentrosymmetric molecular structure. Therefore, TPM can be used for collagen disordering control, which occurs in wound healing [49,50,51]. The SHG-to AF aging index of the dermis (SAAID) is defined as the difference between SHG and AF intensity signals, indicative of type I collagen and elastin, respectively, and normalized to the sum of both signals as follows [52,53,54,55,56]:SAAID = (SHG − AF)/(SHG + AF).(1)

SAAID is a suitable quantitative parameter for characterizing a skin condition, which, as a specific parameter for monitoring wound healing, was proposed in [54], but has not been studied in detail.

This study aims to investigate the dynamics of SAAID in vivo using TPM during LDPDT wound healing by employing the topical administration of two different photosensitizers, 5-ALA and MB, and two laser fluences, 1 J/cm^2^ and 4 J/cm^2^.

## 2. Materials and Methods

### 2.1. An Animal Model of a Wound

In vivo experiments were performed using fifteen male CD1 mice aged 6–7 weeks and weighing 25–30 g according to experimental protocol No.4, 10.02.2021, registration No. 6, as approved by the Bioethical Committee of Tomsk State University. Animals were obtained from the Department of Experimental Biological Models of the Research Institute of Pharmacology, TSC SB RAMS. Before the experiment, the mice were kept for 7 days in the standard conditions of a conventional vivarium with free access to water and food, and a 12/12 light regime in a ventilated room at a temperature of 20 ± 2 °C and a humidity of 60%.

The mice were anesthetized by isoflurane using the Ugo Basile gas anesthesia system, where the mice were put in a glass chamber connected to isoflurane. The parameter on the isoflurane cylinder was set to 3–4%. It is recommended that the mouse be placed inside for 10–15 min. The wound area was depilated by Veet cream (France), rinsed with saline solution, and sterilized with chlorhexidine 20%. No additional drugs were used. The paws’ skin was folded and raised using forceps. Then, the mouse was placed in a lateral position and using medical scissors the skin layers were completely removed and excisional circular wounds (diameter 5 mm) were created, as shown in Figure 1. This procedure was repeated on both the hind paws of each animal. The operations were carried out entirely under the influence of isoflurane gas attached via a mask to the mouse’s nose, but here the isoflurane cylinder was set to 1–1.5%. The experiment was performed in a time-lapsed schedule for wound aging on days 1, 3, 7, and 14. The day of wound formation was denoted as day 0.

### 2.2. Wound Healing Assay

The photos for illustrations were taken using a 16 MP camera with a 26 mm focal length, a 2x magnification lens, and an f/1.9 aperture at observational time points until the wounds healed; examples are presented in Figure 2. The wound size was calculated with a digital caliper by measuring its larger A and minor B diameters on every measurement day. The wound area was determined by the formula S = (A × B × π)/4. The digital caliper accuracy was 0.02 mm. Thus, this value makes an insignificant contribution to the measurement error when calculating the area and it was not considered in our analysis. The percentage of wound closure was calculated as follows:
[(S_0_ − S)/S_0_] × 100,(2)
where S_0_ is the area of the original wound and S is the area of the current wound.

### 2.3. Low Dose Photodynamic Therapy Protocol

The photosensitizers were prepared by dissolving the powder 5-ALA and MB in saline solution. The 0.1–0.2 mL of 5-ALA 20%/MB 0.01% saline solutions were topically administered for 30 min of incubation, dripping directly on the wound when the mouse was placed on a base under isoflurane. The wounds were irradiated by an AlGalnP laser (λ = 630 nm, P = 5 mW) with two fluences: 1 J/cm^2^ and 4 J/cm^2^ for 3 min 45 s and 15 min, respectively. The animals were randomly divided into three basic groups; each group included 5 mice: the control group, LDPDT/1 J/cm^2^ group, and LDPDT/4 J/cm^2^ group. In the LDPDT groups, 5-ALA was applied on the wounds on the right hind paws, MB was applied on the left ones. Thus, in total, there were 5 groups: the control, LDPDT-5-ALA/1 J/cm^2^, LDPDT-MB/1 J/cm^2^, LDPDT-5-ALA/4 J/cm^2^, and LDPDT-MB/4 J/cm^2^ groups. The LDPDT procedure was repeated once immediately after wound formation.

### 2.4. Two-Photon Microscope

The wound was analyzed using a two-photon microscope, MPTflex (Jenlab GmbH, Jena, Germany). The pump laser wavelength was 760 nm and filters at bands 373–387 nm for a SHG signal and at 406–610 nm for an AF signal were used. The repetition frequency was 80 MHz with laser pulse width ~200 fs. The manufacturer’s declared spatial resolution <0.5 μm (horizontal); <2 μm (vertical) with focusing optics: magnification 40× NA 1.3. The object was placed directly under the cover glass of a 100–170 μm thickness. A special metal ring was used as a cover glass holder. The space between the glass and the lens was filled with Carl Zeiss™ Immersol™ immersion oil to obtain a better signal. Skin structure was studied at 0–80 μm depth with a 4 μm step, while the pump laser power was varied from 5 mW at a depth of 0 μm (the beginning of the stratum corneum) to 40 mW at a depth of 80 μm. SHG and AF images were recorded on a 512 × 512-pixel matrix; the image size was 70 × 70 μm. The AF and SHG channels were electronically separated using appropriate spectral filters and recorded in digital matrices in “*.tiff” format in two independent channels. RGB color space was used for visualization, where the SHG and AF signals were shown in the red and green channels, respectively. Five stacks for different skin areas were scanned for each mouse during measurement. AF and SHG images were processed using SPCImage and Python software. This study did not perform data preprocessing as areas with acceptable magnification and image quality were selected. The SAAID index was chosen as the main characteristics of the AF and SHG signals. SAAID had been calculated depending on depth for all groups.

### 2.5. Statistical Analysis

All calculated parameters were expressed as mean ± standard deviation of the mean (SD). The Mann–Whitney U test was used to analyze the differences between two independent datasets. The level of significance was set at *p* < 0.05.

## 3. Results

### 3.1. Visual Observation

The digital photographs of the wounds at different time points were taken for the control and LDPDT groups on the surgery day (0) and successively on days 1, 3, 7, and 14 (Figure 2). The presented photos were only shown to illustrate the process of wound healing in the groups.

During healing, the wound sizes were evaluated according to Equation (2) for five mice from each group (Table 1). The values are presented as mean ± standard deviation. Figure 3 illustrates the wound healing rate for all groups.

### 3.2. In Vivo TPM Imaging

In vivo TPM imaging was used to assess differences in wound healing. Figure 4 shows the SHG and AF images of all groups at a depth of 60 μm. Type I collagen fibers indicated by the SHG signal are shown in red and the elastin fibers indicated by AF signals are in green. The white scale bar shows a length of 10 μm.

Different forms of collagen structure could be observed between days 1 and 14: on day 1, the collagen was disorganized and dispersed in a blurry form. Over the following days, organized collagen gradually started to form, which was confirmed by the increase in SHG signal intensity. On day 14, the collagen fibers were rearranged, cross-linked, and aligned (Figure 4). On the seventh day, the SHG signal intensity in LDPDT groups was higher so the collagen formation was better in LDPDT groups. Figure 5 shows the difference between the organized collagen in healthy tissue and the disorganized collagen in a wound at depths from 4 to 80 µm.

### 3.3. The SAAID Estimation

For quantification of relative amounts of elastin and collagen, the SAAID was estimated for various depths. The corresponding SAAID curve for healthy skin (comparison area) is shown in Figure 6. The SAAID index for the control group is shown in Figure 7a. The SAAID curve started at slightly negative values and was still negative on day 1. On day 3, the curve did not differ much as the values remained negative. On day 7, the SAAID had a minimum value of −0.4 (a depth of 40 μm). On day 14, the curve shape changed compared to day 1 and day 3, the value of SAAID at 70 μm increased to zero, and then the SAAID reached its maximum of 0.2 at a depth of 80 μm. Figure 7b shows the SAAID index for the LDPDT 5-ALA 4 J/cm^2^ group. It was different from the control group; in general the values were larger and on days 7 and 14 they were approximately close to healthy skin values, contrary to the control group. On day 7, the SAAID index started from zero and reached a minimum value of −0.4 (at a depth of 35 μm), rising to zero at 75 μm and a maximum of +0.2 at a depth of 80 μm. With a slight change and larger values, the curve for day 14 is shown. The SAAID reached its minimum at a value of −0.35 (a depth of 25 μm), after which the SAAID curve reached a maximum of +0.3 at a depth of 80 μm. Therefore, the SAAID was found to be substantially affected by the day of the wound healing process. The SAAID of healthy skin vs. control group and LDPDT 5-ALA (1 J/cm^2^ and 4 J/cm^2^) groups on day 14 are shown in Figure 8a. The difference was evident, especially in the range from 40 µm to 80 µm, with higher values for LDPDT 5-ALA/4 J/cm^2^. Additionally, Figure 8b shows the SAAID for healthy skin, the control group, and LDPDT–MB (1 J/cm^2^ and 4 J/cm^2^) groups on day 14.

A comparison of LDPDT 5-ALA and MB/4 J/cm^2^ vs. healthy skin and the control group on day 14 is presented in Figure 9. When MB was employed, the values were closer to those of healthy skin. The specificity of the SAAID for healthy skin, the control group of wound healing, and the LDPDT groups with 5-ALA and MB (1 J/cm^2^ and 4 J/cm^2^) was analyzed. The SAAID differences for the studied groups at various depths are shown in Figure 10 and these differences were more evident at depths of 60 μm and 80 μm.

The SAAID index at depths from 0 to 40 µm on day 14 described a healthy and regenerated skin epidermis layer. The index values for healthy skin, the control area, and skin after LDPDT exposure did not differ significantly, as demonstrated in Figure 7, Figure 8, Figure 9 and Figure 10. It indicates that the index did not show significant differences in the epidermis after recovery or that these changes were minor. There were differences in SAAID values deeper than 60 µm (papillary dermis layer) between the control and LDPDT groups. These differences were most pronounced at a depth of 80 µm. It indicates a significant recovery of the papillary layer during LDPDT so it could be concluded that on day 14 after exposure to LDPDT/4 J/cm^2^, both 5-ALA and MB showed accelerated skin regeneration.

### 3.4. Mann–Whitney U Test

The Mann–Whitney test was applied to assess differences in between the wound skin group and the healthy skin group. SAAID variation *p*-values for the five groups, depths from 40 to 80 μm, in all days are shown in Table 2, Table 3 and Table 4. Statistical power was set at 0.95. The higher the *p*-value, the more minor the difference between a wound skin group and the healthy skin group. Thus, Figure 11 shows the *p*-value on day 14 by depth. This dependence was calculated with a smaller step in depths than the data presented in Figure 10. Indeed, we see that SAAID measured in the 47–50 μm depth interval is the most sensitive to the wound state.

It is worth noting that the *p*-value for LDPDT/4 J/cm^2^ was much higher than for the control group, indicating better healing.

## 4. Discussion

The most common collagens are fibrils (collagens of types I, II, III, V, and XI) and reticular structures (collagens IV, VIII, and X) in the intercellular matrix. Since the type of collagen is directly related to the amino acid sequence underlying its structure, the optical properties of collagens vary depending on its type. As mentioned in the Introduction, collagen I, typical for skin, can be visualized through SHG-microscopy. According to our data, during the wound healing process, especially in the first three days, the collagen was disorganized and disordered. However, after a time, a newly laid collagen matrix gradually formed to fill the wound gap, which was detected by the increase in SHG signal intensity. The disorganized collagen is rearranged, cross-linked, and aligned, which can be observed on day 14 (see Figure 4).

From studying and comparing the five groups (control, LDPDT–5-ALA/4 J/cm^2^, LDPDT–5-ALA/1 J/cm^2^, LDPDT–MB/4 J/cm^2^, and LDPDT–MB/1 J/cm^2^), it has been demonstrated that the 4 J/cm^2^ laser dose is better in comparison with 1 J/cm^2^. For the 4 J/cm^2^ dose, MB enables better healing compared to 5-ALA (see Figure 3). The benefits of MB are its low cost and wide availability. In any case, a fourfold fluence increase did not cause a substantial increase in efficiency, which suggests that at the used concentrations of photosensitizers, it is quite acceptable to choose a fluence between 1 J/cm^2^ and 4 J/cm^2^.

SAAID values have previously been demonstrated for the papillary dermis during skin aging [53]. In [51], SAAID was applied to estimate skin damage by curettage. Every phase of wound healing was studied, as in our research, and similar values were obtained for different depths. Later, the same group of researchers [54] showed similar results for chronic wounds, consistent with our research. The results for estimating wound healing during the photodynamic therapy process using SAAID have not previously been studied in the literature. In this study, variations of the SAAID value during the wound healing process were established. On day 14, in the LDPDT 4 J/cm^2^ group, the SAAID curve was practically the same as for healthy skin (see Figure 7 and Figure 8).

SAAID variance during wound healing demonstrates that the healing efficiency does not increase significantly with a fourfold fluence increase. Moreover, the healing process with low fluence is more uniform across the dermis (see Figure 8).

The *p*-value of the difference of SAAID measured in wounds relative to the healthy skin group was calculated using the Mann–Whitney test. On day 14, the *p*-values were higher, so the differences are smaller relative to healthy skin (see Figure 11). This appears significantly and more clearly in both LDPDT groups. Consequently, SAAID is a suitable quantitative parameter for monitoring the wound healing process.

Interestingly, the most essential SAAID variance between healthy and wounded skin was in the 47–55 μm depth interval (Figure 11). It is an area of the papillary dermis with a large amount of collagen and elastin, but there are practically no capillary blood vessels yet. Recent studies showed that liquid or small molecules preferentially colocalize with elastin fibers [57], affecting its fluorescence in an area of inflammation. Therefore, SAAID change during wound healing can be associated with both collagen and elastin transformation.

## 5. Conclusions

Two-photon microscopy is becoming a conventional tool for noninvasive medical diagnostics of superficial tissues and dynamic monitoring of skin pathology treatment. The latter requires the discovery of specific quantitative criteria of the tissue state. The SAAID index was shown to be a suitable variant of the quantitative criterium for wound healing supervision, including optimal regimes of wound healing low-dose photodynamic therapy.

The development of this approach can be as follows. First of all, fluorescence lifetime imaging combined with SGH and AF channels will give additional information about cell metabolism in a wound area [58]. Other optical modalities, for example, optical coherence tomography, will allow data to be obtained about wound tissue morphology at a lower spatial resolution (the typical value is about 5–7 μm) but with a much larger dimension of tissue area visualization compared to TPM. 

The phototherapy of wounds is closely related to the problem of infectious disease treatment, especially in the presence of antibiotic-resistant bacteria in the bacterial biofilms covering the wound surface [35,59]. Since the development of antibacterial drugs is not keeping pace with microbial resistance, it is necessary to use alternative antibacterial approaches, including antimicrobial PDT. In this sense, the effectiveness of wound healing is determined mainly by the combined effect of light on the bacterial flora hidden in the depth of the wound and the restoration of the main protein components of the skin—collagen and elastin. Both processes require efficient staining of pathogens and dermal cells and a homogeneous distribution of excitation light for the effective production of ROS throughout the entire thickness of the wound.

In this case, it is necessary to overcome the limited depth of light penetration into the tissue in order to enhance the therapeutic effect. It can be done by partially suppressing tissue light scattering when exposed to immersion agents using the optical clearing method [60]. This enables not only an increase in the efficiency of therapy (a decrease in fluence) but also makes it possible to improve optical monitoring of the healing process, providing a greater depth of probing [61], which is vital for deep wounds. It is also important that many optical clearing agents, such as glycerol, are good solvents for photodynamic dyes, ensuring the stability of their absorption spectra [15] and possessing bactericidal properties, which can give synergy in the light treatment of wounds. Deeper photodynamic wound therapy can be achieved by using indocyanine green (ICG) as a PDT dye that is effective in the near-infrared range [16].

## Figures and Tables

**Figure 1 pharmaceutics-14-00287-f001:**
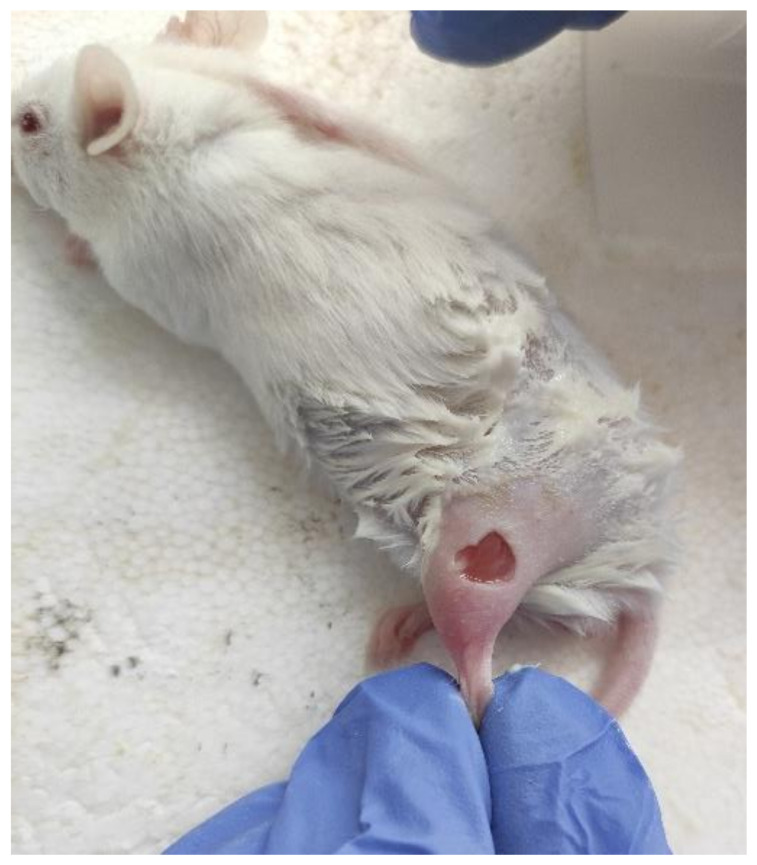
A full-thickness cutaneous wound.

**Figure 2 pharmaceutics-14-00287-f002:**
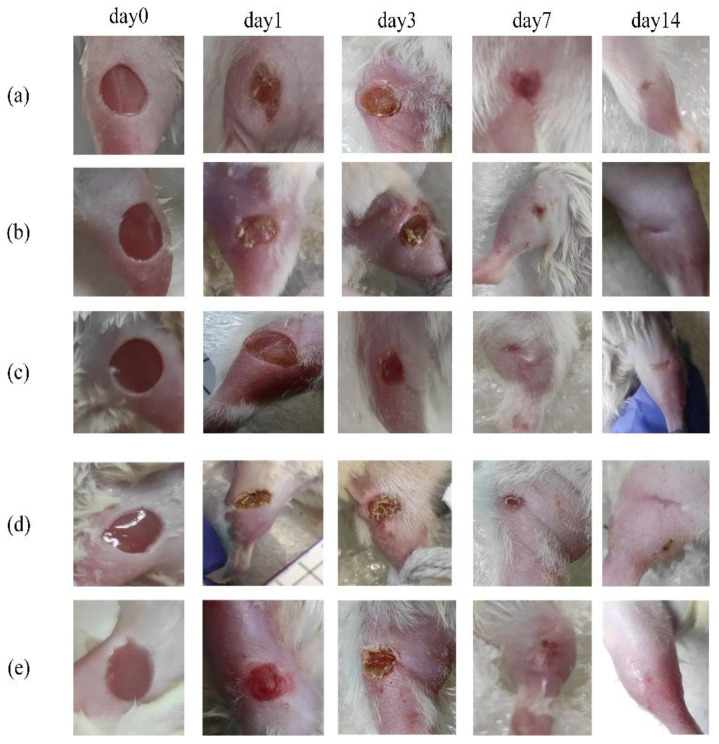
Digital photograph assessment of healing progression from day 0 to day 14: (**a**) control group; (**b**) LDPDT–5-ALA/1 J/cm^2^; (**c**) LDPDT–5-ALA/4 J/cm^2^; (**d**) LDPDT–MB/1 J/cm^2^; (**e**) LDPDT–MB/4 J/cm^2^.

**Figure 3 pharmaceutics-14-00287-f003:**
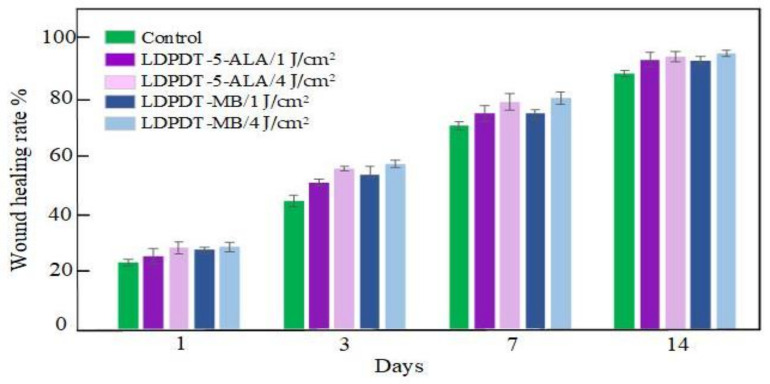
Wound healing rate.

**Figure 4 pharmaceutics-14-00287-f004:**
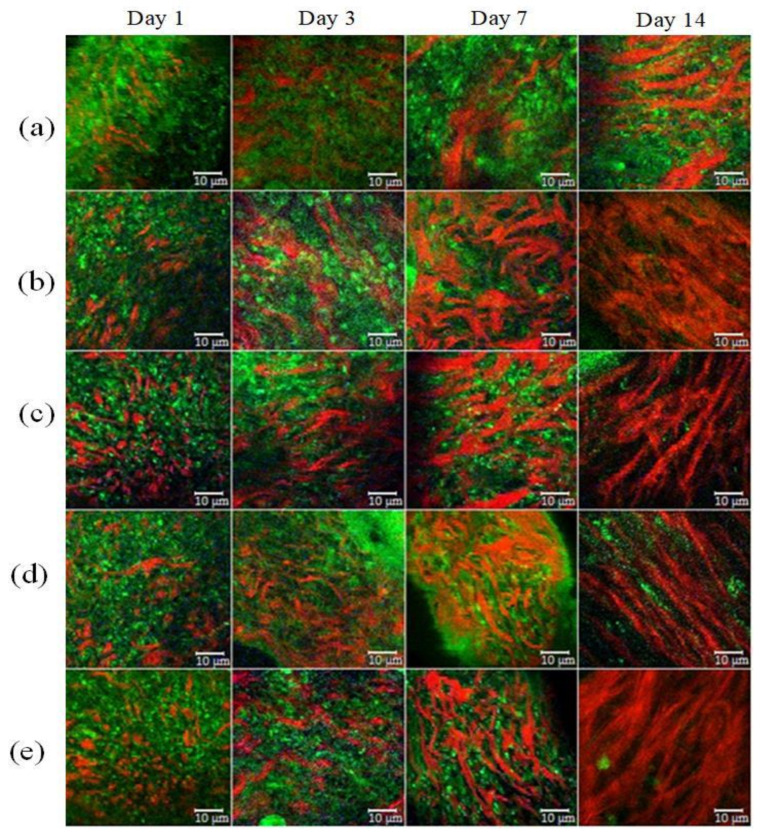
SHG (red) and AF (green) images of wound healing. (**a**) Control group; (**b**) LDPDT–5-ALA/1 J/cm^2^; (**c**) LDPDT–5-ALA/4 J/cm^2^; (**d**) LDPDT–MB/1 J/cm^2^; (**e**) LDPDT–MB/4 J/cm^2^.

**Figure 5 pharmaceutics-14-00287-f005:**
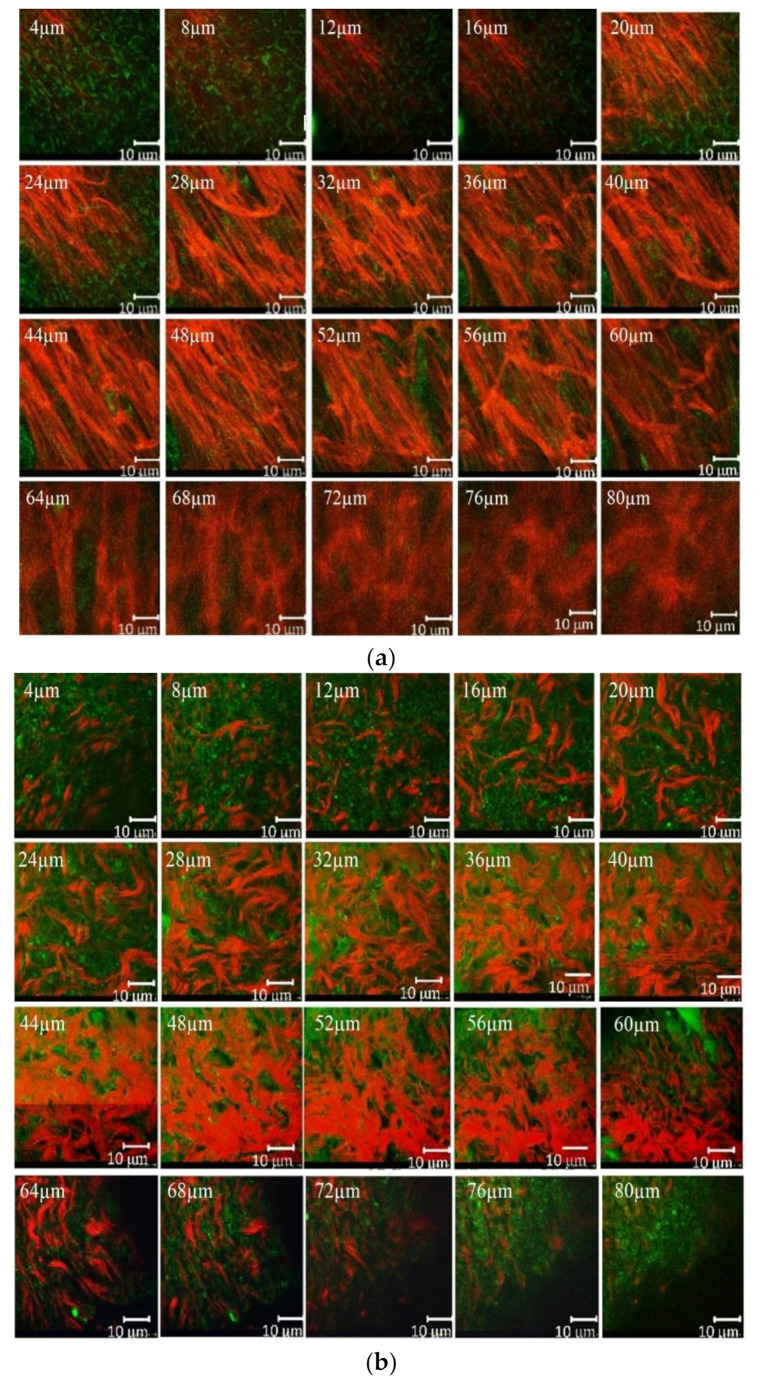
(**a**) SHG (red) and AF (green) of healthy skin at depths from 4 to 80µm; (**b**) SHG (red) and AF (green) of a wound at depths from 4 to 80 µm.

**Figure 6 pharmaceutics-14-00287-f006:**
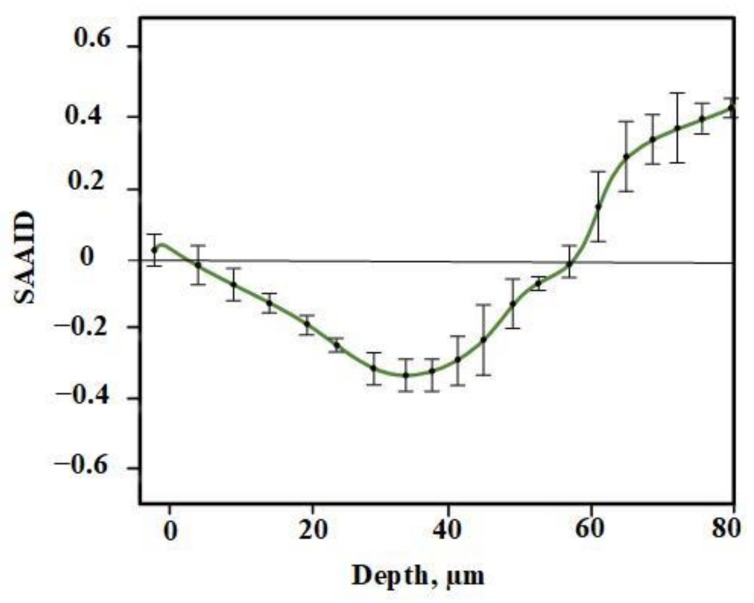
SAAID for healthy skin.

**Figure 7 pharmaceutics-14-00287-f007:**
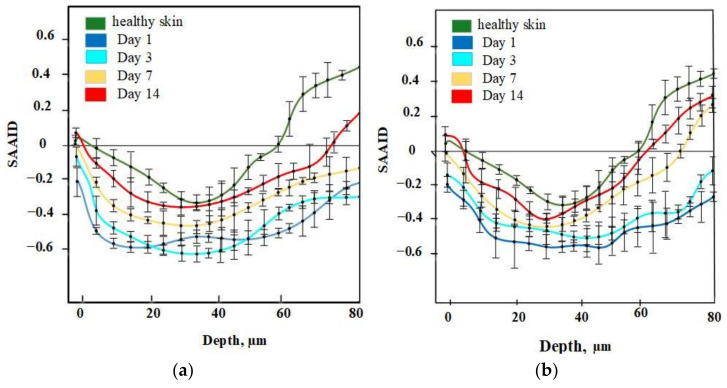
(**a**) SAAID for control group; (**b**) SAAID for LDPDT–5-ALA/4 J/cm^2^ group.

**Figure 8 pharmaceutics-14-00287-f008:**
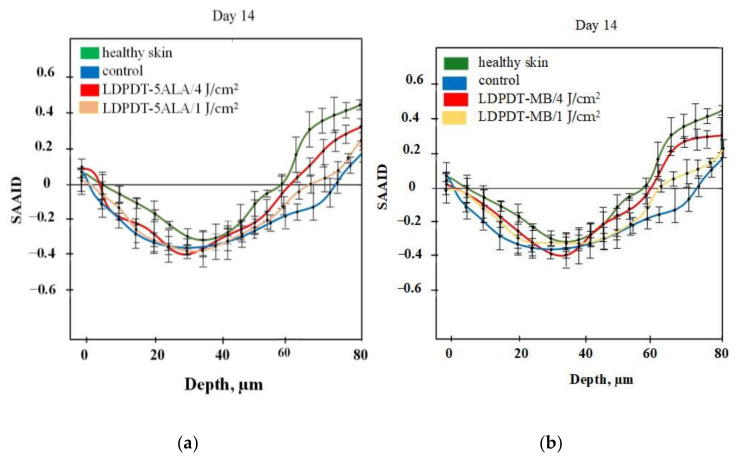
(**a**) SAAID for LDPDT–5-ALA (4 J/cm^2^ and 1 J/cm^2^) groups; (**b**) SAAID for LDPDT–MB (4 J/cm^2^ and 1 J/cm^2^) groups.

**Figure 9 pharmaceutics-14-00287-f009:**
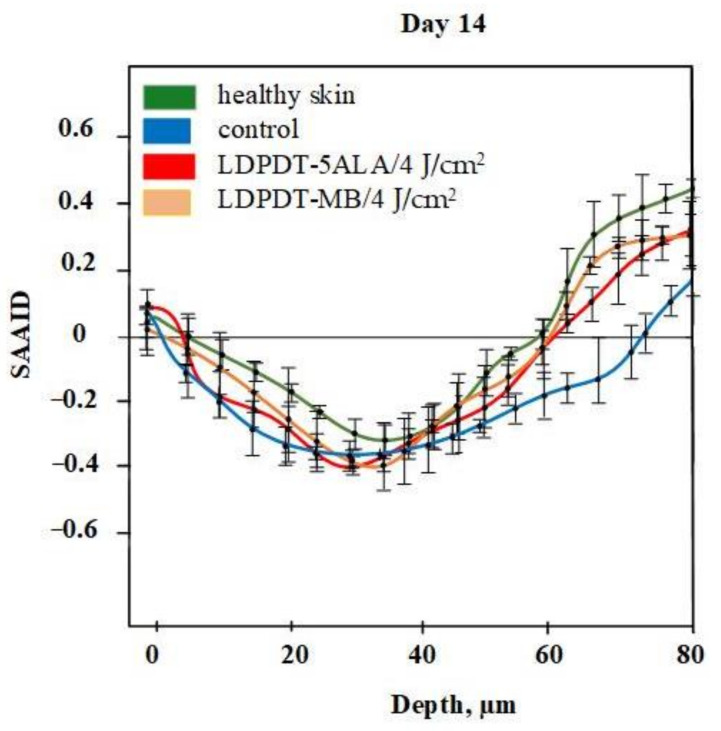
SAAID for LDPDT–5-ALA and LDPDT–MB/4 J/cm^2^ groups.

**Figure 10 pharmaceutics-14-00287-f010:**
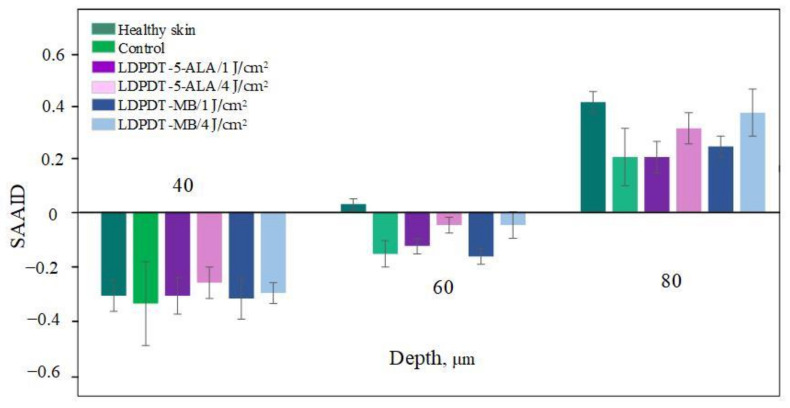
Index SAAID on day 14 at 40 μm, 60 μm and 80 μm for healthy skin, control, LDPDT–5-ALA/1 J/cm^2^, LDPDT–5-ALA/4 J/cm^2^, LDPDT–MB/1 J/cm^2^, and LDPDT–MB/4 J/cm^2^.

**Figure 11 pharmaceutics-14-00287-f011:**
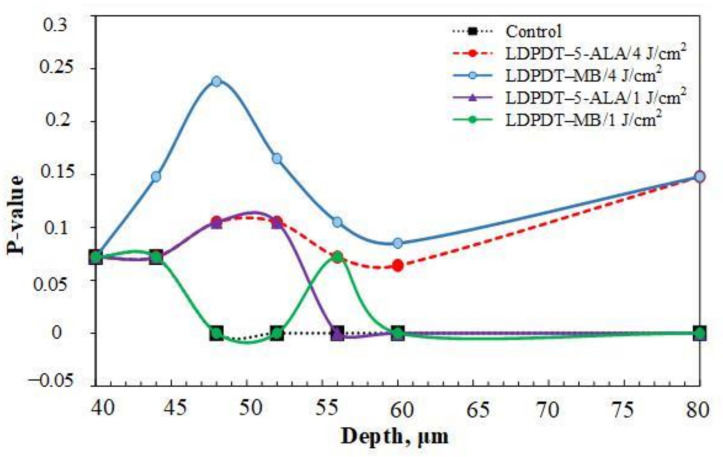
*p*-values of SAAID for all groups at depths from 40 to 60 μm and 80 μm on day 14.

**Table 1 pharmaceutics-14-00287-t001:** The wound size in mm^2^ during wound healing (mean ± standard deviation).

	Day 0	Day 1	Day 3	Day 7	Day 14
control	20.82 ± 1.37	15.89 ± 1.06	11.34 ± 0.88	5.8 ± 0.72	1.98 ± 0.35
LDPDT–5-ALA 1 J/cm^2^	20.02 ± 0.75	14.84 ± 0.89	9.61 ± 0.65	4.71 ± 0.42	0.92 ± 0.14
LDPDT–5-ALA 4 J/cm^2^	20.41 ± 0.54	14.51 ± 0.92	8.79 ± 0.78	3.97 ± 0.57	0.7 ± 0.09
LDPDT–MB 1 J/cm^2^	21.64 ± 0.48	15.53 ± 1.04	9.77 ± 0.43	5.52 ± 0.22	1.05 ± 0.16
LDPDT–MB 4 J/cm^2^	20.02 ± 1.09	14.18 ± 1.11	8.29 ± 0.64	3.62 ± 0.13	0.25 ± 0.07

**Table 2 pharmaceutics-14-00287-t002:** The *p*-value for the SAAID of the control group relative to healthy skin.

*n* = 5	Control			
Depth, μm	Day 1	Day 3	Day 7	Day 14
40	0.0041 *	0.0041 *	0.0061 *	0.072
44	0.0061 *	0.0061 *	0.0061 *	0.072
48	0.0061 *	0.0061 *	0.0061 *	0.030 *
52	0.0061 *	0.0061 *	0.0059 *	0.0061 *
56	0.0061 *	0.0061 *	0.0061 *	0.0061 *
60	0.0041 *	0.0041 *	0.0041 *	0.030 *
80	0.0061 *	0.0061 *	0.0061 *	0.030 *

* *p*  <  0.05.

**Table 3 pharmaceutics-14-00287-t003:** The *p*-value for the SAAID in LDPDT–5-ALA/4 J/cm^2^ and MB/4 J/cm^2^ relative to healthy skin.

*n* = 5	LDPDT–5-ALA/4 J/cm^2^				LDPDT–MB/4 J/cm^2^			
Depth, μm	Day 1	Day 3	Day 7	Day 14	Day 1	Day 3	Day 7	Day 14
40	0.0061 *	0.0061 *	0.0301 *	0.072	0.0061 *	0.0061 *	0.047 *	0.072
44	0.0061 *	0.0061 *	0.0108 *	0.072	0.0061 *	0.0061 *	0.072	0.148
48	0.0061 *	0.0061 *	0.0718	0.105	0.0061 *	0.0061 *	0.202	0.238
52	0.0061 *	0.0061 *	0.105	0.105	0.0061 *	0.0061 *	0.0718	0.165
56	0.0061 *	0.0061 *	0.0108 *	0.072	0.0061 *	0.0061 *	0.0061 *	0.105
60	0.0041 *	0.0041 *	0.0025 *	0.064	0.004 *	0.004 *	0.004 *	0.085
80	0.0059 *	0.0061 *	0.0301 *	0.105	0.0061 *	0.0061 *	0.0718	0.148

* *p*  <  0.05.

**Table 4 pharmaceutics-14-00287-t004:** The *p*-value for the SAAID in LDPDT 5ALA/1 J/cm^2^ and MB/1 J/cm^2^ relative to healthy skin.

*n* = 5	LDPDT–5-ALA/1 J/cm^2^				LDPDT–MB/1 J/cm^2^			
Depth, μm	Day 1	Day 3	Day 7	Day 14	Day 1	Day 3	Day 7	Day 14
40	0.0061 *	0.0061 *	0.0108 *	0.072	0.0061 *	0.0061 *	0.0301 *	0.072
44	0.0061 *	0.0061 *	0.0108 *	0.072	0.0061 *	0.0061 *	0.072	0.072
48	0.0061 *	0.0061 *	0.072	0.105	0.0061 *	0.0061 *	0.047 *	0.0301 *
52	0.0061 *	0.0061 *	0.105	0.105	0.0061 *	0.0061 *	0.047 *	0.006 *
56	0.0061 *	0.0061 *	0.0061 *	0.047 *	0.0061 *	0.0061 *	0.0061 *	0.072
60	0.0041 *	0.0041 *	0.0041 *	0.0025 *	0.0041 *	0.0041 *	0.017 *	0.017 *
80	0.0061 *	0.0061 *	0.047 *	0.047 *	0.0061 *	0.0061 *	0.047 *	0.047 *

* *p*  <  0.05.

## Data Availability

The data presented in this study are available on request from the corresponding author. The data are not publicly available due to privacy or ethical restrictions.

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
