# Peer review of "In Vivo Quantification of the Effectiveness of Topical Low-Dose Photodynamic Therapy in Wound Healing Using Two-Photon Microscopy"

_pharmaceutics, 2022, doi:10.3390/pharmaceutics14020287_

Round 1
Reviewer 1 Report
The manuscript entitled “In vivo quantification of the effectiveness of topical low-dose photodynamic therapy in wound healing using two-photon microscopy"submitted to Pharmaceutics, is an interesting study that aimed to evaluate the effect of low-dose photodynamic therapy on in-vivo wound healing with topical application of 5-aminolevulinic acid and methylene blue.
In my opinion this is a very interesting article.
I would suggest the authors to improve the discussion chapter by including more recent published articles.
I would also suggest the conclusion to be a different chapter of the article providing a precise conclusion.
Author Response
Comments for the Reviewer 1
- I would suggest the authors to improve the discussion chapter by including more recent published articles.
Recent published articles were added to discussion.
- I would also suggest the conclusion to be a different chapter of the article providing a precise conclusion.
The conclusion was added as a different chapter.
Reviewer 2 Report
Cf

Author Response
Comments for the Reviewer 2
Results:
1.Line 130 : why the authors chose male mice? Is there an influence of sex on wound repair?
In this study, male mice were used for more stable results.
2. Line 138 - 144: For animal welfare, did the authors give painkillers at the wound and during the PDT treatment?
All measurements, including wound procedures and PDT treatments, were performed using gas anesthesia isoflurane, without additional anesthetic drugs.
3. Table 1: where are the statistical data: standard errors or SD?
Corrections were added to table 1.
4. Figure 3: where are the standards errors or SD in the graph?
Corrections were added to Figure 3.
Discussion / conclusion:
5. Conclusion and discussion are interesting but a little too short in my humble opinion.
The Conclusion and discussion were separated, rewritten with more details and information.
Reviewer 3 Report
In this manuscript, the authors quantified the effectiveness of topical low-dose photodynamic therapy (LDPDT) in a cutaneous wound mouse model using two-photon microscopy. The manuscript covers a brief overview of LDPDT in wound healing, and reasonable experiment designs to evaluate the effectiveness of the therapy. Overall, this is very logical and integral work. I recommend this manuscript for publication after minor revisions. There are several comments and suggestions:
- In Line 155, please provide more information about topical administration. For example, what’s the weight of cream or gels routinely used on a mouse?
- Are the measured/calculated data of Table 1 and Figure 3 only from only one mouse in Figure 2? If yes, please make a note in figure/table legends; if not, please address how many mice are used, and add standard deviation for the measurement.
- Please revise the figure format in Figure 7─9. There are random lines near treatment labels.
- Please keep the same figure format in Figure 3─10.
- In Line 246, please consider providing a brief discussion about 80 µm depth in Figure 10 to make a smooth transition to Section 3.4 which leads dropping of 80 µm depth statistical analysis.
- In Line 259, “Table 1” should be revised as “Table 3”.
- Please address sample number (n) in Table 2─4.
- In Figure 11, 3D graphics of p-values is not easily and clearly visualized without rotating/zooming the graphics. P-values in Table 2─4 with/without traditional star symbol representations or 2D graphics could be easily read.
Author Response
Comments for the Reviewer 3
-
- In Line 155, please provide more information about topical administration. For example, what’s the weight of cream or gels routinely used on a mouse?
Corrections were added to the manuscript in section “2.3 Low dose photodynamic therapy protocol”.
- Are the measured/calculated data of Table 1 and Figure 3 only from only one mouse in Figure 2? If yes, please make a note in figure/table legends; if not, please address how many mice are used, and add standard deviation for the measurement.
The information about the number of mice was added to table 1, also has been mentioned in sections 2.1 and 2.3. The standard deviation were added to Table 1 and Figure 3.
- Please revise the figure format in Figure 7─9. There are random lines near treatment labels.
Corrections were added to the Figures 7-9.
- Please keep the same figure format in Figure 3─10.
Note taken into account, Figures 3-10 were corrected.
- In Line 246, please consider providing a brief discussion about 80 µm depth in Figure 10 to make a smooth transition to Section 3.4 which leads dropping of 80 µm depth statistical analysis.
A brief discussion was added after fig. 10.
- In Line 259, “Table 1” should be revised as “Table 3”.
“Table 1” was revised as “Table 3”
- Please address sample number (n) in Table 2─4.
The sample number (n) was added to Table 2─4.
- In Figure 11, 3D graphics of p-values is not easily and clearly visualized without rotating/zooming the graphics. P-values in Table 2─4 with/without traditional star symbol representations or 2D graphics could be easily read.
The 3D graphics of p-value were replaced by 2D graphics for all groups on day 14.
Reviewer 4 Report
The manuscript of Hala Zuhayri et al. is a very interesting approach in the context of the low-level light therapy; however, before publication, some concerns of mine should be explained, discussed, or corrected by the authors.
In my opinion, section on the methodology of measurements does not allow these measurements to be reproduced by other researchers because it contains a considerable amount of ambiguity and imprecision. This section should be improved.
It was stated that: „Photos are taken using a digital camera in observation time-points till the wounds heal; examples are presented in Figure 2” (lines 148-149), but there is no information on the standardization for the wound image registration procedure. Realistically, it was impossible to record wound images under stationary conditions (on a live mouse), but the manuscript does not describe whether the recording distance was fixed, what the magnification was and how stability of wound size determination was ensured (after all, depending on the position of the mouse leg, the shape of the wound could change). Furthermore, Fig.2 shows that the images recorded on different days have different magnifications, registration distances, and in some cases there are defocused, so how was the standardization of the wound area determination ensured? This crucial information should be explained in detail in the manuscript.
The manuscript does not say how many mice the study was conducted on. One mouse for each control group? Without precise information on the sample size, it is difficult to assess whether the results obtained are reliable.
There is no information about the objective (and its specification) used in the two-photon microscopy?
There is no information in the manuscript about the technical specifications of the system for capturing wound images, what camera, what lens, what magnification, focal distances (fixed/variable), etc.
The measurement uncertainties are not given in Table 1. This again raises the question of whether sometimes the results presented are not just for a single measurement within each control group. If so, on what basis do the authors conclude that these results are representative?
In the manuscript, it is not directly explained how the wound healing rate was evaluated? Why were there no error bars in Fig.3?
The determination of SAAID is mainly based on the spatial distribution of light intensity of SHG/AF images, but how is it ensured that the registration conditions of these images are standardized? How was a fixed registration distance established? Was any spacer between the sample and the objective used to ensure the fixed imaging distance?
How did the authors separate the different channels of the SHG/AF images to determine the intensity of mean collagen and elastin? By spectral filters or image processing ( RGB channels splitting)?
For the Mann-Whitney test, was the variance in the tested groups compared?
Were the photosensitizers given in the form of an ointment/cream?
The quality of Fig.4 is very poor. Can Authors provide SHG/AF images for TPM in better resolution?
Author Response
Comments for the Reviewer 4
- In my opinion, section on the methodology of measurements does not allow these measurements to be reproduced by other researchers because it contains a considerable amount of ambiguity and imprecision. This section should be improved.
The section of methods was improved.
- It was stated that: „Photos are taken using a digital camera in observation time-points till the wounds heal; examples are presented in Figure 2” (lines 148-149), but there is no information on the standardization for the wound image registration procedure. Realistically, it was impossible to record wound images under stationary conditions (on a live mouse), but the manuscript does not describe whether the recording distance was fixed, what the magnification was and how stability of wound size determination was ensured (after all, depending on the position of the mouse leg, the shape of the wound could change). Furthermore, Fig.2 shows that the images recorded on different days have different magnifications, registration distances, and in some cases there are defocused, so how was the standardization of the wound area determination ensured? This crucial information should be explained in detail in the manuscript.
The photos in Figure 2 are only to explain and illustrate the process of wound healing in the five groups. To make it more obvious, we would add information about this to the text in the 3.1 section. The length and width of the wound were measured using digital calipers on all measurement days. The wound area was estimated by length and width using the formula from section 2.2. The status of tissue was estimated using a multi-photon microscope and its techniques: second harmonic generation, autofluorescence, and the SAAID index.
- The manuscript does not say how many mice the study was conducted on. One mouse for each control group? Without precise information on the sample size, it is difficult to assess whether the results obtained are reliable.
The number of mice was mentioned in section “2.1 An animal model of wound”, and the number in each group was added in section “2.3 Low dose photodynamic therapy protocol”.
- There is no information about the objective (and its specification) used in the two-photon microscopy?
Additional information regarding optics and laser were provided in section 2.4.
- There is no information in the manuscript about the technical specifications of the system for capturing wound images, what camera, what lens, what magnification, focal distances (fixed/variable), etc.
The information about the camera were added to section ”2.2 Wound healing assay”
- The measurement uncertainties are not given in Table 1. This again raises the question of whether sometimes the results presented are not just for a single measurement within each control group. If so, on what basis do the authors conclude that these results are representative?
The results in Table 1 are the mean of 5 measurements for five mice, the standard deviation was added in table 1 and figure 3.
- In the manuscript, it is not directly explained how the wound healing rate was evaluated?
To make it more obvious, we add additional information in section 2.2. Wound healing rate (the percentage of wound closure) was calculated as follows: [(S0-S)/S0]×100,where S0 is the area of the original wound, and S is the area of the actual wound, as mentioned on line 151.
- Why were there no error bars in Fig.3?
Corrections were added to the figure 3.
- The determination of SAAID is mainly based on the spatial distribution of light intensity of SHG/AF images, but how is it ensured that the registration conditions of these images are standardized? How was a fixed registration distance established? Was any spacer between the sample and the objective used to ensure the fixed imaging distance?
For ensuring that the standardized registration conditions of these images we used CE-certified medical tomography which has clear specification, which was included in the text. The object was placed directly under the cover glass, with thickness described in the text. A special metal ring was used as a cover glass holder. The distance between the glass and the lens was filled with emissive oil (see text) to obtain a better signal.
- How did the authors separate the different channels of the SHG/AF images to determine the intensity of mean collagen and elastin? By spectral filters or image processing ( RGB channels splitting)?
Additional information has been added to the captions for Figures 4 and 5. The AF/SHG channels were electronically separated using appropriate spectral filters and recorded in digital matrices in tiff format in two independent channels (the details were also included in the text). For visualization, the RGB space was used, where the SHG signal was selected as the red channel, and the AF signal was selected as the green channel.
- For the Mann-Whitney test, was the variance in the tested groups compared?
Mann-Whitney test was carried out for every wound group in comparison with healthy skin group, for 5 measurements for each sample. The P-value was calculated using the standard formula: U_1=R_1-(n_1 (n_1+1))/2 and U_2=R_2-(n_2 (n_2+1))/2, where n1 and n2 are the sample size for sample 1 and 2 respectively, and R1, R2 are the sum of the ranks in sample 1 and 2 respectively.
- Were the photosensitizers given in the form of an ointment/cream?
This information has been added to the section ”2.3 Low dose photodynamic therapy protocol section”.
- The quality of Fig.4 is very poor. Can Authors provide SHG/AF images for TPM in better resolution?
Better quality of SHG/AF images was provided (see Figure 4 and 5).
Reviewer 5 Report
Dear authors,
this well-designed article, but I have some suggestions:
- you did not specify the sample calculation method and there is no power analysis, please add
- regarding Results, the results are vaguely written, it is necessary to briefly explain each statement in the table in a simple sentence so that the reader is completely clear what the result is before looking at the table or figure, as you described TPV imaging results
- the results should be written in the past tense, please correct
- the discussion part should be re-written, you should compare your one results with the results of other authors previously publish and try to comment why there are or there are not some differences
- please write the concise conclusion in the separate part
- please provide the confirmation of your proof-reading or professional English editing
Author Response
Comments for the Reviewer 5
- you did not specify the sample calculation method and there is no power analysis, please add
The result part has been expanded taking into account this remark.
The P-value for Mann Whitney test was calculated using the standard formula: p = ||z||, where z-score was calculated as z = (max(u1, u2)- mean rank) / sd. U_1=R_1-(n_1 (n_1+1))/2 and U_2=R_2-(n_2(n_2+1))/2, where n1 and n2 are the sample size for sample 1 and 2 respectively , and R1, R2 are the sum of the ranks in sample 1 and 2 respectively, .
Power analysis was estimated as the probability of rejecting the main hypothesis when the competing hypothesis is true. Indirectly, P-value allows you to estimate the probability of making a Type II error. In our case, with a confidence of 0.05, it is clear which differences are significant and which are not (power = 0.95).
The analysis of the sample size was limited because of the complexities of the experiment. From the data presented, it can be seen that the spread of values in data is not large, which allows us to speak about obtained results confidence.
- Regarding Results, the results are vaguely written, it is necessary to briefly explain each statement in the table in a simple sentence so that the reader is completely clear what the result is before looking at the table or figure, as you described TPV imaging results
More details and information were written in results.
- the results should be written in the past tense, please correct
Note taken into account, results were rewritten in past tense.
- the discussion part should be re-written, you should compare your one results with the results of other authors previously publish and try to comment why there are or there are not some differences.
Comparing results with previous studies was added to conclusion.
- please write the concise conclusion in the separate part
The Conclusion and discussion were separated, rewritten with more details and information.
- please provide the confirmation of your proof-reading or professional English editing
We had recruited Mitchell Peter, Doctor of Pedagogy, Full member of the Institute of Linguists of Great Britain, for professional editing.
Round 2
Reviewer 4 Report
The authors' explanations and corrections are sufficient. In my opinion, the manuscript in its present form is suitable for publication, although editorial corrections related to citation of references are necessary.